# West Nile Virus Lineage 1 in Italy: Newly Introduced or a Re-Occurrence of a Previously Circulating Strain?

**DOI:** 10.3390/v14010064

**Published:** 2021-12-30

**Authors:** Giulia Mencattelli, Federica Iapaolo, Federica Monaco, Giovanna Fusco, Claudio de Martinis, Ottavio Portanti, Annapia Di Gennaro, Valentina Curini, Andrea Polci, Shadia Berjaoui, Elisabetta Di Felice, Roberto Rosà, Annapaola Rizzoli, Giovanni Savini

**Affiliations:** 1Istituto Zooprofilattico Sperimentale dell’Abruzzo e del Molise, 64100 Teramo, Italy; f.iapaolo@izs.it (F.I.); f.monaco@izs.it (F.M.); o.portanti@izs.it (O.P.); a.digennaro@izs.it (A.D.G.); v.curini@izs.it (V.C.); a.polci@izs.it (A.P.); s.berjaoui@izs.it (S.B.); e.difelice@izs.it (E.D.F.); g.savini@izs.it (G.S.); 2Center Agriculture Food Environment, University of Trento, 38098 Trento, Italy; roberto.rosa@unitn.it; 3Fondazione Edmund Mach, San Michele all’Adige, 38098 Trento, Italy; annapaola.rizzoli@fmach.it; 4Istituto Zooprofilattico Sperimentale del Mezzogiorno, 80055 Napoli, Italy; giovanna.fusco@izsmportici.it (G.F.); claudio.demartinis@izsmportici.it (C.d.M.)

**Keywords:** Arbovirus, WNV, WNV-L1, Italy, surveillance, whole genome sequencing, phylogenetic analysis

## Abstract

In Italy, West Nile virus (WNV) appeared for the first time in the Tuscany region in 1998. After 10 years of absence, it re-appeared in the areas surrounding the Po River delta, affecting eight provinces in three regions. Thereafter, WNV epidemics caused by genetically divergent isolates have been documented every year in the country. Since 2018, only WNV Lineage 2 has been reported in the Italian territory. In October 2020, WNV Lineage 1 (WNV-L1) re-emerged in Italy, in the Campania region. This is the first occurrence of WNV-L1 detection in the Italian territory since 2017. WNV was detected in the internal organs of a goshawk (*Accipiter gentilis*) and a kestrel (*Falco tinnunculus*). The RNA extracted in the goshawk tissue samples was sequenced, and a Bayesian phylogenetic analysis was performed by a maximum-likelihood tree. Genome analysis, conducted on the goshawk WNV complete genome sequence, indicates that the strain belongs to the WNV-L1 Western-Mediterranean (WMed) cluster. Moreover, a close phylogenetic similarity is observed between the goshawk strain, the 2008–2011 group of Italian sequences, and European strains belonging to the Wmed cluster. Our results evidence the possibility of both a new re-introduction or unnoticed silent circulation in Italy, and the strong importance of keeping the WNV surveillance system in the Italian territory active.

## 1. Introduction

West Nile virus (WNV) is a mosquito borne single-stranded RNA virus, a member of the Japanese encephalitis (JE) serocomplex belonging to the genus Flavivirus within the Flaviviridae family [1]. WNV is maintained in nature through an endemic cycle which involves mosquitoes (Diptera; *Culicidae*) as vectors, and birds as reservoir hosts [2]. Humans and horses are considered “dead-end” hosts: they may develop disease; however, they are not able to infect vectors, and maintain the virus in the environment [3]. To date, eight different lineages of WNV have been described [4]. Lineages 1 and 2, often associated with cases of encephalitis in humans and horses, are by far those most widespread in Europe and the Mediterranean basin [5,6,7,8].

In Europe, WNV lineage 1 (WNV-L1) circulation was first evidenced in the 1960s in France, Portugal, and Cyprus [8,9,10]. Thirty years later, cases associated to WNV-L1 infection were reported in humans and horses in North African, Western, and Eastern European countries [8]. The WNV-L1 strains responsible for the Morocco (1996), Italy (1998), Israel (1998), and France (2000) human and horse cases [11,12,13,14,15,16] grouped into the Western-Mediterranean (WMed) clade, whereas those responsible for the Romanian (1996) and Russian (1999) human cases clustered in the Eastern-European clade [17,18,19]. A strong relationship has been observed between the WNV-L1 Israeli and Northern African strains which emerged in the Mediterranean region in the late 1990s, and the ones responsible for the 1999 New York epidemics, suggesting a viral flow between North Africa and North America via the Middle East [20]. In the early 2000s, WNV-L1 outbreaks were again recorded in Morocco, France, and Romania [16,21,22,23]. Starting from 2005, however, WNV lineage 2 (WNV-L2) strains belonging to the Hungarian and Volgograd clades started spreading to some South-Eastern and Eastern European countries [24,25,26,27,28,29,30]. In some regions, co-circulation of both WNV-L1 and L2 were observed [8,24,29,31,32,33]. Following the WNV-L2 incursions and spread, the circulation of WNV-L1 strains was less frequently observed. From 2010, it was reported in Morocco, Algeria, Tunisia, Spain, and Portugal [34,35,36,37,38,39]. Eastern WNV-L1 strains instead kept circulating in Romania, and spread in Bulgaria and Ukraine [27,40,41]. Human and horse infections with Eastern WNV-L1 strains were also reported in Turkey in 2010 and following years [42,43]. In recent years, WNV-L1 circulation appeared to be limited to Northern African countries, the Iberian Peninsula, Cyprus, Turkey, Israel, and Serbia [36,37,38,39,44,45,46,47,48,49,50].

In Italy, WNV-L1 emerged for the first time in the Tuscany region in 1998 [12,13]. Since then, the Italian Ministry of Health implemented a national veterinary surveillance plan for monitoring WNV in areas at risk of viral introduction and circulation [32]. The surveillance system did not detect any relevant WNV circulation until 2008, when WNV-L1 was identified in mosquitoes, birds, horses, and humans in the area surrounding the Po River delta [51]. Since 2008, WNV epidemics caused by genetically divergent isolates have been registered every year [29,52]. The phylogenetic analysis of the isolates confirmed the hypothesis of the virus overwintering, and the endemization in local host populations [53,54]. Between 2010 and 2011, WNV-L1 circulation further spread into Southern Italy, involving Sicily, Apulia, Calabria, Basilicata, and Sardinia regions [32,52,55]. Partial genome sequencing showed that strains isolated in the same area in 2011 were almost identical, but divergent from those responsible for the outbreak in Northern Italy in 2008–2009 [56,57] which, conversely, appeared strictly related to the WNV strains circulating in Europe and Israel from late 2004 to 2011 [55]. In 2011, WNV-L2 was first reported in Italy [31,58,59]. Since then, WNV-L1 circulation was reported only sporadically in birds and mosquito pools from North-Eastern regions (2012–2014, 2017) and Sardinia (2015–2016) [32]. In Italy, WNV-L1 was last detected in a mosquito pool collected in the Piacenza province in 2017 (https://westnile.izs.it/j6_wnd/wndItalia;jsessionid=D0C9EB639E7C322D0EFC34ECEB8E4D8E, accessed on 14 September 2021). According to a risk-based ranking of the Italian provinces, wild birds, mainly corvids (Eurasian jay, *Garrulus glandarius;* Carrion crow, *Corvus corone;* and Magpie, *Pica pica*), poultry, horses, and mosquitoes, are constantly sampled to obtain an early detection of WNV circulation, and reduce the risk of human transmission. To date, 16 out of the 20 Italian regions are considered endemic (Italian epidemiological reports).

In October 2020, the WNV-L1 strain re-emerged in Italy, notably in the Campania region. Within the wildlife monitoring plan of the Campania region (PGMFS) and the National Plan for Prevention, Surveillance, and Response to Arbovirus 2020–2025, WNV-L1, was detected in two wild birds, a kestrel (*Falco tinnunculus*) and a goshawk (*Accipiter gentilis*), found in Naples and Caserta provinces, respectively. This is the first occurrence of WNV-L1 detection in migratory and resident raptor birds since 2018.

In this paper, we describe the two cases, and characterize by whole genome sequencing (WGS) and phylogenetic analysis the WNV-L1 responsible for the goshawk infection.

## 2. Materials and Methods

### 2.1. Bird Conditions and Laboratory Analyses

In October 2020, two wild birds, a kestrel and a goshawk, were found in critical conditions in Naples (Somma Vesuviana municipality 40.878958, 14.426694) and Caserta (Trentola Ducenta municipality 40.976368, 14.1664) provinces, respectively (Figure 1). The two birds, immediately transferred to the Regional Center for Wild Animals (CRAS) for rescue operations, died 48 h after the transfer. In detail, at the time of admission at the clinic, the kestrel was in a comatose state showing mydriasis. The radiography also evidenced the fracture of the tibia, and the presence of two bullet fragments. At the time of acceptance, the goshawk showed head injury, head tilt, and right lower limb ataxia. Necropsies were carried out, and selected tissues (heart, kidney, spleen, and brain) were collected, pooled, and homogenized in a sterile phosphate-buffered saline (PBS). Viral RNA was extracted from 200 μL supernatants using Qiasymphony^®^ DSP automatic instrumentation (Germantown, MD, USA) according to the manufacturer’s instructions. Quantitative reverse transcription polymerase chain reactions (qRT PCR) to detect WNV-L1 and/or -L2 RNA was performed at the U.O.C. Virology of IZSM as described by Del Amo and colleagues [60]. The samples were sent to the National Reference Centre for Foreign Animal Diseases (CESME) at the Istituto Zooprofilattico Sperimentale of Abruzzo and Molise in Teramo (IZSAM) for WNV confirmation and further analysis.

### 2.2. Virus Strain and Laboratory Tests (IZSAM)

Virus RNA of the two samples’ homogenates was extracted at IZSAM by using the MagMAX CORE Nucleic Acid Purification KIT (Applied Biosystem, Thermo Fisher Scientific, Life Technologies Corporation, TX, USA) according to the manufacturer’s instructions. The virus RNA was tested by two qRT-PCR: (i) a 1-step RT- PCR assay for the simultaneous detection of WNV-L1 and 2 strains, by using the QuantiTect Probe RT-PCR Kit (QIAGEN) [60]; and (ii) an RT-PCR assay for detection of all known lineages of West Nile virus [61], by using the Superscript III Platinum OneStep qRT-PCR System (Invitrogen). The RNA detected in the goshawk tissue samples was further fully sequenced by using next generation sequencing (NGS) technology. Briefly, total RNA was treated with TURBO DNase (Thermo Fisher Scientific, Waltham, MA, USA) at 37 °C for 20 min, and then purified by an RNA Clean & Concentrator™-5 Kit (Zymo Research, Irvine, CA, USA). The purified RNA was used for the assessment of sequencing independent single primer amplification protocol (SISPA) [62,63]. In detail, viral RNA underwent cDNA synthesis using 200 units of the SuperScript^®^ IV Reverse Transcriptase (Thermo Fisher Scientific, Waltham, MA, USA), in the presence of 5X SSIV Buffer, 50 μM of the random hexamer FR26RV-N 5′-GCCGGAGCTCTGCAGATATCNNNNNN-3′, 10 mM of dNTPs mix, 100 mM of DTT, and 40U of RNAse OUT RNase inhibitor (Thermo Fisher Scientific, Waltham, MA). The reaction was incubated at 23 °C for 10 min, and 50 °C for 50 min. After an inactivation step at 80 °C for 10 min, 12.5 Units of 3′-5′ Klenow Polymerase (New England Biolabs, Ipswich, MA, USA) were directly added to the reaction to perform the second strand cDNA synthesis. The incubation was carried out at 37 °C for 1 h, and 75 °C for 10 min. Next, 5 μL of ds cDNA were added to 45 μL of PCR master mix containing 5X Q5 Reaction Buffer, 10 mM of dNTPs, 40 μM of the random primer FR20 Rv 5′-GCCGGAGCTCTGCAGATATC-3′, Q5^®^ High Fidelity DNA polymerase (NEB, New England Biolabs, Ipswich, MA, USA), and Q5 High Enhancer [64]. The reaction was incubated at 98 °C for 10 s, 65 °C for 30 s, 72 °C for 3 min, and 72 °C for 2 min. The PCR product was purified using the Molecular Biology Kit BioBasic (Biobasic inc., Markham, ON, Canada), and then quantified by using the Qubit^®^ DNA HS Assay Kit (Thermo Fisher Scientific, Waltham, MA, USA). The sample was diluted to obtain a concentration of 100–500 ng, and used for library preparation by using the Illumina DNA Prep kit (Illumina Inc., San Diego, CA, USA) according to the manufacturer’s protocol. Deep sequencing was performed on the NextSeq 500 (Illumina Inc., San Diego, CA, USA) using the NextSeq 500/550 Mid Output Reagent Cartridge v2, 300 cycles, and standard 150 bp paired end reads. FASTQ files were generated using NextSeq Reporter (Illumina). The sequencing run delivered 300 Mb of sequence data. The reads obtained were trimmed using a Trimmomatic script (Trimmomatic v0.36) to remove low quality and short reads [65]. Furthermore, the reads were quality controlled by using FastQC v0.11.5 [63,66]. The resulting 2,149,990 reads were de novo assembled using SPADES v3.11.1 [67]. Based on genome assemblies, a de novo filtering for the scaffolds with a minimum length of 200 nucleotides, and a matching for the best reference for each assembly using ABRicate was carried out [63]. Finally, a mapping with the references found in the previous step using Bowtie2 (v.2.1.0) was performed [68]. The length of the final assembly (GenBank accession number MW627239) was of 10,990 bp, and it showed 98.33% nucleotide identity to the reference sequence FJ483548.

### 2.3. Phylogenetic Analyses

A total of 64 sequences with information on country and year of isolation were downloaded from Genbank or from the IZSAM database for this study. In particular, the study was conducted using 34 complete genome sequences, 18 polyprotein gene complete coding DNA sequences (cds), and 12 partial cds (polyprotein and envelope glycoprotein gene). Sequences were obtained from mosquitoes (n = 7), birds (n = 24), horses (n = 6), humans (n = 16), and mice (n = 1) (10 sequences were from unknown hosts). They were selected from Italy (n = 27), Portugal (n = 2), France (n = 4), Spain (n = 4), Romania (n = 2), Israel (n = 2), Morocco (n = 2), Nigeria (n = 1), Russia (n = 5), Senegal (n = 3), Cyprus (n = 1), Turkey (n = 1), Japan (n = 1), Americas (n = 7), Kenya (n = 1), and Central African Republic (CAR) (n = 1). With the addition of 1 new sequence, a total of 65 sequences were aligned using the ClustalW algorithm implemented in Ugene v. 37.0 software (available at http://ugene.net/download.html, accessed on 12 August 2021). Aligned sequences were manually curated using BioEdit v. 7.2 software (available at http://www.mbio.ncsu.edu/BioEdit/bioedit.html, accessed on 12 August 2021). Metadata of all sequences referred to in this manuscript can be found in Table 1.

The Bayesian phylogenetic analysis was performed through Bayesian Inference (BI) using a general time-reversible with gamma-distributed rate variation, a gamma category count of 4, and an invariant sites model (GTR + Γ + I), as selected by Akaike’s information criterion (AICc) in jModelTest 0.1 [69]. A Bayesian MCMC approach using BEAST with JRE v2.6.3 was then employed. Ten independent MCMC runs with up to 100 million generations were performed to ensure the convergence of estimates. Tracer v.1.7.1 (available at http://beast.bio.ed.ac.uk/Tracer, accessed on 12 August 2021) was used to ensure convergence during MCMC by reaching effective sample sizes greater than 100. Trees were summarized in a maximum clade-credibility tree with common ancestor heights after a 10% burn-in [69] using TreeAnnotator v.2.6.3. A maximum likelihood tree was estimated using FigTree v1.4.4 [70] after identical alignment and curating methods. FigTree was run using the GTR + Γ + I nucleotide model with 2000 Γ-rate categories, exhaustive search settings, with 5000 bootstrap replications using the Shimodaira–Hasegawa (SH) test. All sequence alignments referred to in this manuscript can be found in Appendix A.

## 3. Results

### 3.1. Strain Characterization and Phylogenetic Analysis

#### 3.1.1. WNV-L1 Detection

WNV-L1 was detected and confirmed in both the kestrel and the goshawk organs (heart, kidney, spleen, and brain pooled together in PBS).

#### 3.1.2. Phylogenetic Tree Inferred with Maximum-Likelihood Analysis

Phylogenetic inference of WNV using a maximum-likelihood tree is shown in Figure 2 and Figure 3. All Shimodaira–Hasegawa values are displayed at respective nodes.

According to the analysis, WNV-L1 is represented by three main clusters: (1) the Mediterranean-Eastern European-Kenyan WNV cluster, that includes isolates from Senegal (1993), Kenya (1998), Romania (1996–97), Russia (1999, 2000, 2006), and Cyprus (2016); (2) the WMed cluster, including isolates from Europe (Italy 1998, 2008, 2009, 2011 and 2020; Spain 2007, 2008; Portugal 2004 and 2009; France 2003, 2004 and 2015), Israel 2000–2007, and Morocco 1996–2003; and (3) the Israeli-American WNV cluster, including several American strains, isolated between 1999 and 2002 (all phylogenetically very similar among them), in addition to strains from Japan (2007), Europe (France 2003 and Spain 2007), and Africa (Nigeria 1965, CAR 1967, Senegal 1979, and Tunisia 2011), confirming a WNV clusterization already highlighted in the past [71,72] (Figure 2 and Figure 3).

The genome analysis includes the new strain TE_362447_2020 (accession number: MW627239 lineage 1, Italy, 2020) into the WMed single monophyletic group (Figure 3).

A close similarity is observed between the MW627239 and the 2008–2011 groups of sequences clustered separately into the WMed subtype (Figure 3). Phylogenetically, a similarity is also observed between the Italian WNV-L1 (2008, 2011, 2020) and the viral strains circulating in Europe in the recent past (Spain 2010, France 2015), although they are less temporally close than the Italian strains [71].

#### 3.1.3. Full-Length Polyprotein Sequencing

Using BioEdit v. 7.2 software (available at http://www.mbio.ncsu.edu/BioEdit/bioedit.html, accessed on 12 August 2021), an amino acid sequence comparison between MW627239 and representative WNV-L1 Italian, European, and American isolates was performed. In particular, the strains (i) 115803 (accession number: FJ483548, Italy, 2008); (ii) WNV Italy 1998-equine (accession number: AF404757, Italy, 1998); (iii) Italy/2008/M-203204 (accession number: JF719066, Italy, 2008); (iv) Italy/2008/J-242853 (accession number: JF719065, Italy, 2008); (v) 21412 (accession number: MW835362, Italy, 2011); (vi) Italy/2013/Livenza/35.1 (accession number: KF647253, Italy, 2013); (vii) Spain/2010/H-1b (accession number: JF719069, Spain, 2010); (viii) GE-2o/V (accession number: FJ766332, Spain, 2007); (ix) Akela/France/2015 (accession number: MT863559, France, 2015); and (x) NY99 (accession number: NC009942, USA, 1999) were included in the analysis. Only representative amino acid residues were analyzed [7,52,73,74,75]. Results are shown in Table 2.

Using Blast (https://blast.ncbi.nlm.nih.gov/Blast.cgi, accessed on 12 August 2021), a nucleotide and amino acid pairwise identity analysis was conducted among MW627239 and the Italian, European, and American sequences listed above, as shown in Table 3. Among the Italian representative sequences belonging to the WNV-1 WMed single monophyletic group in the maximum-likelihood tree, the average nucleotide and amino acid pairwise identity was evidenced to be 98.13% (s.d. = 0.36) and 99.73 % (s.d = 0.07), respectively.

Results evidence a close genetic relatedness of the WNV-L1 strain that re-emerged in the Campania region in 2020, and the Italian and European strains belonging to the WNV-L1 WMed sub-cluster.

## 4. Discussion

This paper reports the first evidence of WNV-L1 strain circulation in the Campania region, Italy. The strain was detected in October 2020 in two wild birds found moribund in nearby areas, only a few days apart. It was the first detection of WNV-L1 after several years. The last WNV-L1 strain circulation evidenced in Italy dates back to 2017, when it was found in a pool of mosquitoes from Northern Italy (https://westnile.izs.it/j6_wnd/wndItalia;jsessionid=D0C9EB639E7C322D0EFC34ECEB8E4D8E, 14 September 2021).

The last evidence of WNV-L1 circulation in an area most nearby the Campania region dates back to 2016 when the circulation of a WNV-L1 strain was responsible for the death of several wild birds in Sardinia [31]. The first question which clearly came to mind when tackling this finding was: “was this strain the result of a new introduction or was it just a re-occurrence of a strain already circulating?”.

The first assumption supposes the virus probably extinguished and reintroduced through migratory birds. The species where the strains were detected did not help in clearing the question. Goshawks and kestrels can in fact be considered either migrant or resident birds. Unfortunately, it was not possible to check the carcasses, and eventually make out the bird behavior from wing, claw, and feather characteristics [74,75,76]. In support of the re-introduction of WNV-L1 is the lack of detection for consecutive years by the national surveillance program, which has been in place since 2002, and has been drawn to detect virus circulation early. In the same way, a phylogenetic relationship (%), and nucleotide and amino acid similarities are observed between the new sequence, MW627239, and some European sequences obtained in the past (Spain 2010, France 2015) (Figure 3, Table 2 and Table 3). This might suggest a possible viral circulation in the Mediterranean followed, by a re-introduction in Italy in 2020.

Still on the phylogenic tree, however, the new strain MW627239 shows high nucleotide sequence identity (%) with the 2008–2011 Italian sub-clusters of the WMed single monophyletic group (Figure 3). In this respect, the second hypothesis, the WNV-L1 re-occurrence, seems to be the most credible scenario. Even if, in all these years, WNV has repeatedly proven its ability to overwinter and become endemic in many Italian regions, silence periods are factually not unusual for WNV-L1 [53,71]. After its first occurrence in the Tuscany region in 1998, WNV re-appeared in the areas surrounding the Po River delta after 10 years [55]. Similarly, in Sardinia, WNV-L1 was not detected for 3 consecutive years between 2011 and 2015 (https://westnile.izs.it/j6_wnd/wndItalia;jsessionid=D0C9EB639E7C322D0EFC34ECEB8E4D8E, 14 September 2021). In this particular case, the WNV-L1 presence was accidentally uncovered in two wild birds when they were in critical conditions. The presence of clinical signs in birds would indeed facilitate the discovery of virus circulation on many occasions, as observed in several WNV epidemics of the early 2000s [16,76,77]. Even though the majority of the WNV infections in birds are usually mild or asymptomatic [78], some species, such as birds of prey, jays, and crows, are highly susceptible, and can develop severe and even fatal encephalitis [78,79,80]. Clinical symptoms associated to WNV-L1 infection have been mainly reported in the orders of Passeriformes (corvids, blue jays, magpies) and Falconiformes (birds of prey) [81,82,83]. Fatal infections have been described in European eagles in Spain [84,85], and geese and poultry in Hungary [24]. The two WNV-L1 infected birds found in Campania were in critical conditions. Though the comatose state of the kestrel was likely the consequence of a gunshot wound, the origin of the clinical picture observed in the goshawks is not easy to assess. Head injury, head tilt, and right lower limb ataxia are definitely signs of neurologic pathology. However, whether these symptoms were a consequence of the head injury or vice versa couldn’t be determined. In nature, goshawks have been shown to be highly susceptible to WNV infection, probably because of their predatory habits [81]. Oral transmission of WNV by feeding on infected prey has been described, and is believed to be an important route of transmission in birds of prey [79,83,86]. Both WNV-L1 and L2 experimental infections conducted in American kestrels (*Falco sparverius*), golden eagles (*Aquila chrysaetos*), red-tailed hawks (*Buteo jamaicensis*), barn owls (*Tyto alba*), and great horned owls (*Bubo virginianus*) showed high level of viraemia, and important clinical symptoms, such as lethargy, inappetence, body weight loss, and muscle tremor [82,87,88]. However, looking back over the past 13 years of circulation in Italy, WNV-L1 seems to be unable to cause important clinical manifestations and deaths in birds [52].

The virulence of WNV depends on several factors related to the pathogen, hosts, and their interaction [89,90]. Concerning the pathogen, changes in the amino acid positions may significantly influence the WNV strain virulence [7]. Multiple genetic variations correlated with increased or decreased pathogenicity have been highlighted in genetic and phenotypic studies of WNV mutants [74]. Among the WNV genes, the NS3 helicase domain is considered a virulence determinant [52,91]. In particular, increased avian virulence due to the point mutation NS3-T249P has been reported in American crows (*Corvus brachyrhynchos*) and in site-directed mutagenesis experiments [73,92,93,94]. In support of our analysis, the Italian strain MW627239 is not characterized by the NS3-T249P point mutation. A threonine residue was observed at the 249 position in the strain MW627239, as well as in JF719065 and MW835362, circulating in Italy in 2008 (jay) and 2011 (owl), and in JF719069 and MT863559, circulating in Spain (horse) and France (human) in 2010 and 2015, respectively. Furthermore, the amino acid valine at the residue 159 of the E protein is considered a determinant of WNV neurovirulence, influencing viral replication and pathogenesis, and being involved in WNV infection and T-cell infiltration in the brain [75]. This amino acid is observed in the isolate NC009942, circulating in the USA in 1999, but not in the Italian and European strains, all characterized by isoleucine at this residue position. This point mutation might also help explain the low pathogenesis observed among birds in Italy and, more generally, in Europe. It is likely that the WNV pathogenicity is the result of a complex series of events which involve the virus, the vectors, and the hosts. Further studies correlated with the WNV genotype and phenotype may help in understanding the mechanisms underlying WNV clinical signs in birds, and the emergence of new pathogenic phenotypes.

## 5. Conclusions

In recent years, WNV-L2 has been, by far, the most frequent lineage detected in Italy, whereas WNV-L1 was detected only occasionally. Since 2008, about 1500 wild birds and 5000 resident birds belonging to target species (carrion crow, magpie, Eurasian jay) have been annually tested by real time RT-PCR to monitor the circulation of WNV L1 and L2 strains in Italy. Among them, 326 wild birds and 804 target species were found positive to WNV (https://westnile.izs.it/j6_wnd/home, 16 December 2021). Concerning WNV-L1, it was last detected in a sparrow hawk (*Accipiter nisus*) and in two carrion crows in the Sardinia region during the 2016 vector season.

Serological analysis conducted in Italy on humans and horses between 2008 and 2020 identified anti-WNV-IgM in 1189 persons and 1196 horses. No evidence of WNV circulation was detected in the Campania region [95] (https://westnile.izs.it/j6_wnd/home, 16 December 2021). The detection of the WNV-L1 strain in two wild birds described in this study emphasizes the importance of having in place an efficient surveillance system, and, in particular, the early warning function played by some avian species in detecting WNV circulation.

## Figures and Tables

**Figure 1 viruses-14-00064-f001:**
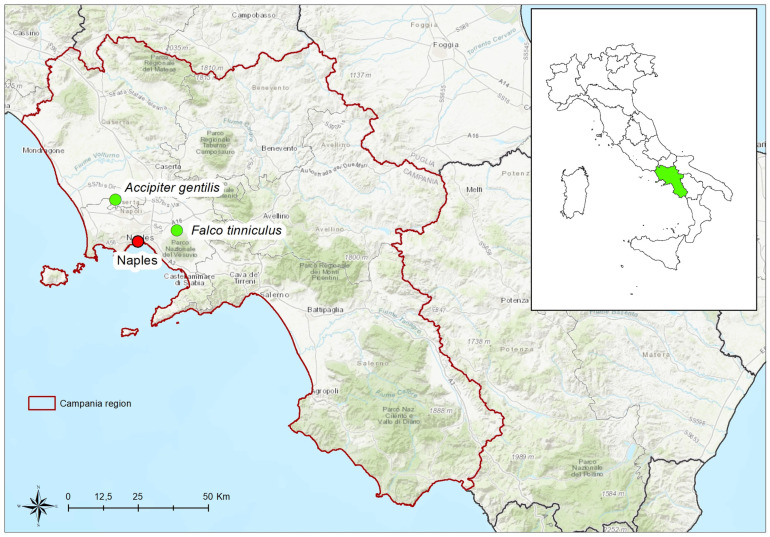
Geolocation of sites where the two wild birds were found.

**Figure 2 viruses-14-00064-f002:**
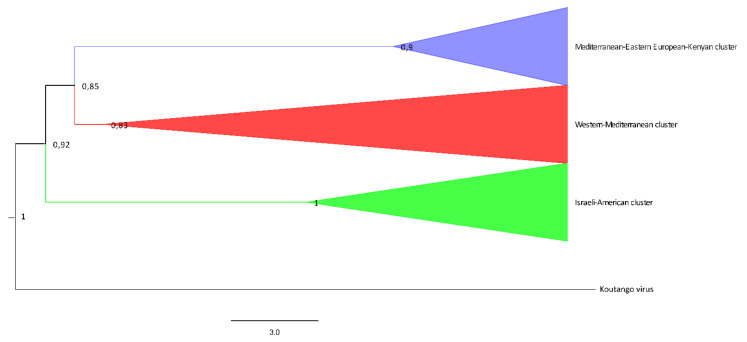
Maximum likelihood phylogenetic tree of the WNV complete and partial genome sequences analyzed in this study. Violet, red, and green triangles represent the Mediterranean-Eastern European-Kenyan subtype, the Western Mediterranean subtype, and the Israeli-American subtype of WNV sequences, respectively. The Koutango virus strain EU082200 has been chosen as outgroup. The tree with the highest log-likelihood is shown. The Bayesian phylogenetic analysis was performed through Bayesian Inference (BI) using a general time-reversible with gamma-distributed rate variation, a gamma category count of 4, and an invariant sites model (GTR + Γ + I). The evolutionary distances were computed using the optimal GTR + Γ + I model, with 2000 Γ-rate categories and 5000 bootstrap replications using the Shimodaira–Hasegawa (SH) test. The percentage of successful bootstrap replicate (n5000) is indicated at nodes. A Bayesian MCMC approach using BEAST with JRE v2.6.3 was employed. Ten independent MCMC runs with up to 100 million generations were performed to ensure the convergence of estimates. Tracer v.1.7.1 was used to ensure convergence during MCMC by reaching effective sample sizes greater than 100. Trees were summarized in a maximum clade-credibility tree with common ancestor heights after a 10% burn-in using TreeAnnotator v2.6.3.

**Figure 3 viruses-14-00064-f003:**
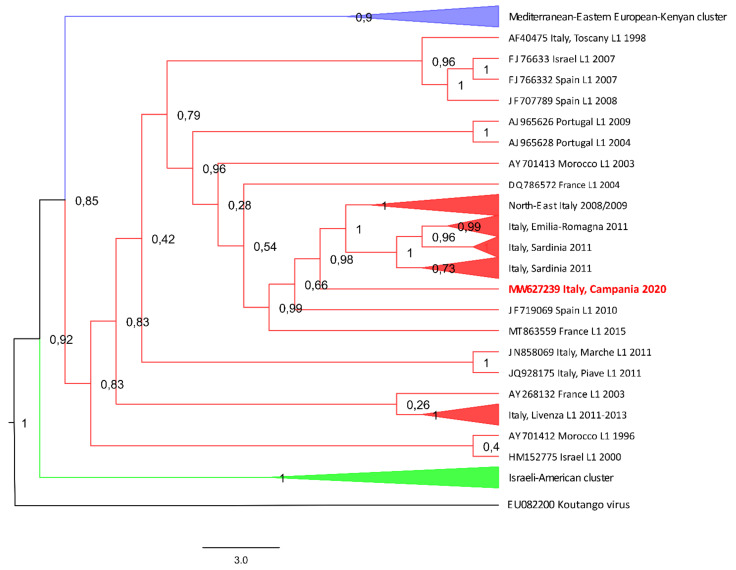
Maximum likelihood phylogenetic tree of the WNV complete and partial genome sequences analyzed in this study. A detailed version of the phylogenetic tree is shown, showing all the Western Mediterranean WNV sequences. GenBank accession numbers are indicated for each strain, with country, lineage, and year of isolation. The WNV-L1 strain TE_362447_2020, obtained from the goshawk found in the Campania region in October 2020, is highlighted in red. The Koutango virus strain EU082200 has been chosen as outgroup. The tree with the highest log-likelihood is shown. The Bayesian phylogenetic analysis was performed through Bayesian Inference using a general time-reversible with gamma-distributed rate variation, a gamma category count of 4, and an invariant sites model (GTR + Γ + I). The evolutionary distances were computed using the optimal GTR + Γ + I model, with 2000 Γ-rate categories and 5000 bootstrap replications using the Shimodaira–Hasegawa (SH) test. The percentage of successful bootstrap replicate (n5000) is indicated at nodes. A Bayesian MCMC approach using BEAST with JRE v2.6.3 was employed. Ten independent MCMC runs with up to 100 million generations were performed to ensure the convergence of estimates. Tracer v.1.7.1 was used to ensure convergence during MCMC by reaching effective sample sizes greater than 100. Trees were summarized in a maximum clade-credibility tree with common ancestor heights after a 10% burn-in using TreeAnnotator v2.6.3.

**Table 1 viruses-14-00064-t001:** Relevant data regarding the isolates used for the present study.

Strain Number	Viral Species	Isolation Material	Host	Country	Year of Isolation	Accession Number
TE.362447.2020	WNV L1	Homogenate	Northern goshawk	Italy	2020	MW627239
TE.15803.2008	WNV L1	-	Magpie	Italy	2008	FJ483548
TE.15217.2008	WNV L1	-	Magpie	Italy	2008	FJ483549
TE.229892.2008	WNV L1	-	Magpie	Italy	2008	KU573077
Ita09	WNV L1	Blood	Human	Italy	2009	GU011992
Italy/2009/J-225677	WNV L1	C636 cells P1, Vero cells P3	Eurasian jay	Italy	2009	JF719068
Italy/2009/FIN	WNV L1	-	Human	Italy	2009	KF234080
Italy/2008/J-242853	WNV L1	C636 cells P1, Vero cells P3	Eurasian jay	Italy	2008	JF719065
Italy/2009/G-223184	WNV L1	C636 cells P1, Vero cells P3	Gull	Italy	2009	JF719067
Italy/2008/M-203204	WNV L1	C636 cells P1, Vero cells P3	Magpie	Italy	2008	JF719066
204913/2009	WNV L1	-	*Culex pipiens* mosquito	Italy	2009	KU573078
TE.14444.2011	WNV L1	Organ pool	Magpie	Italy	2011	MW835356
TE.17196.2011	WNV L1	Organ pool	Owl	Italy	2011	MW835357
TE.17208.2011	WNV L1	Organ pool	Crow	Italy	2011	MW835358
TE.20224/1.2011	WNV L1	Plasma	Chicken	Italy	2011	MW835359
TE.21370.2011	WNV L1	Plasma	Horse	Italy	2011	MW835361
TE.20875.2011	WNV L1	Organ pool	Eurasian jay	Italy	2011	MW835360
TE.23237.2011	WNV L1	Plasma	Chicken	Italy	2011	MW835363
TE.20224/8.2011	WNV L1	Plasma	Chicken	Italy	2011	Under publication
TE.21412.2011	WNV L1	Brain	Owl	Italy	2011	MW835362
04.05	WNV L1	Brain	Horse	Morocco	2003	AY701413
PT5.2	WNV L1	-	-	Portugal	2004	AJ965628
PT6.16	WNV L1	-	-	Portugal	2009	AJ965626
Spain/2010/H-1b	WNV L1	Brain	Horse	Spain	2010	JF719069
WN Italy 1998-equine	WNV L1	-	Equine	Italy	1998	AF404757
96-111	WNV L1	Brain	Equine	Morocco	1996	AY701412
PaAn001	WNV L1	-	-	France	2003	AY268132
France 405/04	WNV L1	Brain	House sparrow	France	2004	DQ786572
WNV_0304h_ISR00	WNV L1	-	Human	Israel	2000	HM152775
GE-2o/V	WNV L1	Vero cells	Golden eagle	Spain	2007	FJ766332
GE-1b/B	WNV L1	BSR cells	Golden eagle	Israel	2007	FJ766331
HU6365/08	WNV L1	-	*Culex perexiguus* mosquito	Spain	2008	JF707789
RO97-50	WNV L1	-	*Culex pipiens* mosquito	Romania	1996	AF260969
KN3829	WNV L1	-	*Culex univittatus* mosquito	USA	2003	AY262283
VLG-4	WNV L1	Brain	Human	Russia	1999	AF317203
Tomsk/bird/2006/A4	WNV L1	-	Blyth’s reed warbler	Russia	2006	MN149538
LEIV-Vlg99-27889	WNV L1	Brain	Human	Russia	1999	AY277252
LEIV-Vlg00-27924	WNV L1	Blood	Human	Russia	2000	AY278442
Italy/2012/Livenza/37.1	WNV L1	Urine	Human	Italy	2012	KC954092
Italy/2012/Livenza/31.1	WNV L1	Culture viral isolate from blood	Human	Italy	2012	JX556213
Italy/2013/Livenza/35.1	WNV L1	Plasma	Human	Italy	2013	KF647253
Italy/2011/Livenza	WNV L1	Plasma	Human	Italy	2011	JQ928174
Italy/2013/Livenza/37.1	WNV L1	Urine	Human	Italy	2013	KF823807
Akela/France/2015	WNV L1	-	Human	France	2015	MT863559
Italy/2011/AN-1	WNV L1	Urine	Human	Italy	2011	JN858069
Italy/2011/Piave	WNV L1	Urine	Human	Italy	2011	JQ928175
WNV_Cy2016	WNV L1	Urine	Human	Cyprus	2016	MF797870
T2	WNV L1	Blood	Equine	Turkey	2011	KJ958922
ArB310/67	WNV L1	-	-	Central African Republic	1967	GQ851608
IBAN7019	WNV L1	-	-	Nigeria	1965	GQ851607
ArD27875	WNV L1	-	Mosquito	Senegal	1979	GQ851606
PaH001	WNV L1	-	-	France	2003	AY268133
NY99-flamingo382-99	WNV L1	Chicken embryo	Flamingo	USA	1999	AF196835
ABB-B13	WNV L1	-	Mouse	Spain	2007	KC407667
NY99	WNV L1	Vero cell P2	-	USA	1999	NC009942
NY99iso-1	WNV L1	Vero cell E6	-	Japan	2007	FJ411043
NY99-crow-V76/1	WNV L1	-	American crow	USA	1999	FJ151394
3356K VP2	WNV L1	Kidney	American crow	USA	2000	EF657887
WNV-1/US/BID-V6527/2001	WNV L1	Kidney and Spleen	American crow	USA	2001	KJ501343
WNV-1/US/BID-V6506/2002	WNV L1	Kidney and Spleen	American crow	USA	2002	KJ501489
Ast99-901	WNV L1	Blood	Human	Russia	1999	AY278441
KN3829	WNV L1	-	*Culex univittatus* mosquito	Kenya	1998	AF146082
RO97-50	WNV L1	-	-	Romania	1997	AF130362
SEN-ArD93548	WNV L1	-	Mosquito	Senegal	1993	AF001570
Dak Ar D 5443	KOUTANGO VIRUS	-	-	Senegal	2013	EU082200

Relevant data regarding the isolates used for the present study.

**Table 2 viruses-14-00064-t002:** Amino acid sequence comparison of MW627239 (goshawk, Italy, 2020) and representative WNV-L1 Italian, European, and American strains, conducted using BioEdit v. 7.2 software. The strains (i) 115803 (accession number: FJ483548, Italy, 2008); (ii) WNV Italy 1998-equine (accession number: AF404757, Italy, 1998); (iii) Italy/2008/M-203204 (accession number: JF719066, Italy, 2008); (iv) Italy/2008/J-242853 (accession number: JF719065, Italy, 2008); (v) 21412 (accession number: MW835362, Italy, 2011); (vi) Italy/2013/Livenza/35.1 (accession number: KF647253, Italy, 2013); (vii) Spain/2010/H-1b (accession number: JF719069, Spain, 2010); (viii) GE-2o/V (accession number: FJ766332, Spain, 2007); (ix) Akela/France/2015 (accession number: MT863559, France, 2015); and (x) NY99 (accession number: NC009942, USA, 1999) were included in the analysis. Only representative amino acid residues were analyzed. C protein: amino acid (aa) 1–105; PreM protein: aa 124–290; M protein: aa 216–290; E protein: aa 291–791; NS1 protein: aa 792–1143; NS2A protein: 1114–1374; NS2B protein: 1375–1505; NS3 protein: 1506–2124; NS4A protein: 2125–2250; NS4B protein: 2274–2528; NS5 protein: 2529–3433. Significant amino acid residue substitutions are highlighted in red.

Viral Protein	Amino Acid Position	MW627239 ITA 2020	AF404757 ITA 1998	JF719066 ITA 2008	FJ766332 SPA 2007	JF719065 ITA 2008	FJ483548 ITA 2008	MW835362 ITA 2011	KF647253 ITA 2013	NC_009942 USA 1999	JF719069 SPA 2010	MT863559 FRA 2015
C	34	M	M	M	** V **	M	M	M	M	M	M	M
	100	S	S	S	S	S	S	S	S	S	S	S
prM	72	S	S	S	S	S	S	S	S	S	S	S
M	36	I	I	I	I	I	I	I	I	I	I	I
E	35	S	S	S	S	S	S	S	S	S	S	S
	51	** T **	A	A	** T **	A	A	A	A	A	A	A
	76	T	T	T	T	T	T	T	T	T	T	T
	88	P	P	P	** S **	P	P	P	P	P	P	P
	126	T	T	T	T	T	T	T	T	** I **	T	T
	153	G	G	G	G	G	G	G	G	G	G	G
	159	I	I	I	I	I	I	I	I	** V **	I	I
	278	T	T	T	T	T	T	T	T	T	T	T
	312	L	L	L	L	L	L	L	L	L	L	L
	442	V	V	V	V	V	V	V	V	V	V	V
NS1	17	S	S	S	S	S	S	S	S	S	S	S
	35	Y	Y	Y	** H **	Y	Y	Y	Y	Y	Y	Y
	45	I	I	I	I	I	I	I	I	I	I	I
	70	S	S	S	S	S	S	S	S	** A **	S	S
	94	E	E	E	E	E	E	E	E	E	E	E
	138	P	P	P	P	P	P	P	P	P	P	P
	141	K	K	K	K	K	K	K	K	K	K	K
	188	V	V	V	V	V	V	V	V	V	V	V
	208	D	D	D	** H **	D	D	D	D	D	D	D
	288	S	S	S	S	S	S	S	S	S	S	S
	289	E	E	E	** G **	E	E	E	E	E	E	E
NS2A	85	I	I	** V **	I	** V **	** V **	I	I	I	I	I
	104	N	N	N	N	N	N	N	N	N	N	N
	119	H	H	H	H	H	H	H	H	H	H	H
	128	E	E	E	E	E	E	E	E	E	E	E
	138	V	V	V	V	V	V	V	V	V	V	V
	165	G	G	G	G	G	G	G	G	G	G	G
NS2B	82	D	D	D	D	D	D	D	D	D	D	D
	83	G	G	G	G	G	G	G	G	G	G	G
	103	A	A	A	A	A	A	A	** V **	** V **	A	A
	120	V	** I **	V	V	V	** I **	V	V	V	V	V
NS3	46	F	F	F	F	F	F	F	F	F	F	F
	244	Q	Q	Q	Q	Q	Q	Q	Q	Q	Q	** H **
	249	T	T	T	** P **	T	** P **	T	** P **	** P **	T	T
	356	I	I	I	I	I	I	I	I	** T **	I	I
	496	L	L	L	L	L	L	L	L	L	L	L
	503	N	N	N	N	N	N	N	N	N	N	N
	521	D	D	D	D	D	D	D	D	D	D	D
NS4A	85	V	V	** I **	V	** I **	** I **	V	V	** A **	V	V
	100	P	P	** S **	P	** S **	** S **	P	P	P	P	P
	122	P	P	P	P	P	P	P	P	P	P	P
NS5	53	H	H	H	H	H	H	H	H	H	H	H
	54	P	P	P	P	P	P	P	P	P	P	P
	257	D	D	D	D	D	D	D	D	D	D	D
	258	V	V	** A **	V	** A **	** A **	** A **	V	V	V	V
	280	K	K	K	K	K	K	K	K	K	K	K
	372	V	V	V	V	V	V	V	V	V	V	V
	374	Y	Y	Y	Y	Y	Y	Y	Y	Y	Y	Y
	422	R	R	K	R	K	K	K	R	R	R	R
	426	E	E	E	A	E	E	E	E	E	E	E
	436	M	M	M	I	M	M	M	M	M	M	M
	526	T	T	T	T	T	T	T	T	T	T	T
	653	F	F	F	F	F	F	F	F	F	F	F
	681	T	T	T	T	T	T	T	T	T	T	T

**Table 3 viruses-14-00064-t003:** Nucleotide versus amino acid similarities of representative WNV Italian, European, and American strains. Pairwise identity analyses have been conducted among MW627239 and other WNV L-1 Italian, European, and American sequences, using Blast. Among the Italian representative sequences belonging to the WNV-1 WMed single monophyletic group in the maximum-likelihood tree, the average nucleotide and amino acid pairwise identity was evidenced to be 98.13% (s.d. = 0.36) and 99.73% (s.d = 0.07), respectively.

	MW627239 ITA 2020	AF404757 ITA 1998	JF719066 ITA 2008	JF719065 ITA 2008	FJ483548 ITA 2008	MW835362 ITA 2011	KF647253 ITA 2013	NC_009942 USA 1999	FJ766332 SPA 2007	JF719069 SPA 2010
MW627239 ITA 2020	-	98.5%	98.31%	98.33%	98.33%	98.02%	97.67%	95.80%	97.96%	98.27%
AF404757 ITA 1998	99.85%	-	98.50%	98.53%	98.55%	98.18%	98.54%	96.44%	98.93%	98.51%
JF719066 ITA 2008	99.68%	99.77%	-	99.95%	99.41%	98.93%	98.13%	96.04%	98.37%	98.75%
JF719065 ITA 2008	99.71%	99.80%	99.91%	-	99.94%	98.95%	98.15%	96.07%	98.41%	98.78%
FJ483548 ITA 2008	99.68%	99.83%	99.88%	99.91%	-	98.93%	98.15%	96.10%	98.39%	98.76%
MW835362 ITA 2011	99.68%	99.77%	99.71%	99.74%	99.71%	-	97.85%	95.76%	98.00%	98.43%
KF647253 ITA 2013	99.74%	99.83%	99.65%	99.68%	99.71%	99.65%	-	96.03%	98.34%	98.11%
NC_009942 USA 1999	99.62%	99.71%	99.56%	99.59%	99.62%	99.53%	99.71%	-	96.29%	96.16%
FJ766332 SPA 2007	99.65%	99.68%	99.50%	99.53%	99.56%	98.75%	99.62%	99.50%	-	98.35%
JF719069 SPA 2010	99.65%	99.74%	99.56%	99.59%	99.56%	99.39%	99.62%	99.50%	99.48%	-

## Data Availability

Sequence data are available via NCBI. The accession numbers for the sequences used can be found in Table 1.

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
