# Peer review of "West Nile Virus Lineage 1 in Italy: Newly Introduced or a Re-Occurrence of a Previously Circulating Strain?"

_viruses, 2021, doi:10.3390/v14010064_

Round 1

Reviewer 1 Report

Manuscript ID: viruses-1485392

Title: West Nile virus lineage 1 in Italy: a newly introduced or a re-occurrence of a previously circulating strain?

The manuscript describes the detection of WestNile virus, Lineage 1 re-emergence after 3 years in Italy. The virus was confirmed from internal organs of two birds and sequenced. A detailed genome analysis was performed and a phylogenetic evaluation was performed evaluating the possibility of re-introduction or silent circulation of the virus strain. The detailed genome analysis of the European WNV-L1 strains is performed indicating clustering of the Mediterranean sequences. It produces interesting results, which highlight the importance of continuous monitoring in the area of interest. The article is well written and the analysis is performed in a correct way. I recommend the manuscript for publications.

Minor comments:

  • Do you have the data on how many birds are screened annually? If in the years, where no WNV-L1 was confirmed in the hosts significantly less birds are screened, this could produce a bias, and would confirm the silent circulation of the virus.
  • Abstract, line 19: WNV was isolated… - Did you isolate the virus? There is no description in the methods. The methods include the description of direct sequencing from tissue homogenates.

Reviewer 2 Report

General comments:

the manuscript is well written and clear for the most parts. In some instances, I suggest to split long sentences and rephrase some, as I specified in the specific comments. The authors conducted a very detailed and comprehensive genomic analysis, including amino acid comparison to sequences of related WNV strains. Since the entire data is based on two sequences from two clinical cases, it would have been helpful to add some more data on the infected birds from which the RNA was extracted. If no pathological data, photos or histological images are available, it would be helpful to include information on the tissue from which the RNA was extracted.

A second general comment pertains to the possible reasons for the re-occurrence of WNV L1 after a few years of apparent absence. The authors performed a very extensive literature survey

 (95 citations), which is more appropriate for a review, rather than a research article. However, there was very little information on cased of horses and humans, with respect to circulating WNV in Italy and in the Napoli region in particular, in recent years. Such information could have been helpful in hypothesizing what are the circumstances under which L1 re-emerged. I think that in light of the extensive literature screening, one or two sentences that put the study’s finding in the context with horses and human infections (if such data exist), could further emphasize the importance of the work.
